# Mixture Risk Assessment of Complex Real-Life Mixtures—The PANORAMIX Project

**DOI:** 10.3390/ijerph192012990

**Published:** 2022-10-11

**Authors:** Beate I. Escher, Marja Lamoree, Jean-Philippe Antignac, Martin Scholze, Matthias Herzler, Timo Hamers, Tina Kold Jensen, Marc Audebert, Francois Busquet, Dieter Maier, Michael Oelgeschläger, Maria João Valente, Henriette Boye, Sebastian Schmeisser, Gaud Dervilly, Matteo Piumatti, Soléne Motteau, Maria König, Kostja Renko, Maria Margalef, Ronan Cariou, Yanying Ma, Andreas Frederik Treschow, Andreas Kortenkamp, Anne Marie Vinggaard

**Affiliations:** 1Department of Cell Toxicology, Helmholtz Centre for Environmental Research—UFZ, DE-04318 Leipzig, Germany; 2Environmental Toxicology, Department of Geoscience, Eberhard Karls University Tübingen, DE-72076 Tübingen, Germany; 3Department Environment & Health, Faculty of Science, Vrije Universiteit Amsterdam, 1081 HV Amsterdam, The Netherlands; 4LABERCA, Oniris, INRAE, 44307 Nantes, France; 5Centre for Pollution Research and Policy, Environmental Sciences Division, Brunel University London, Kingston Lane, Uxbridge UB8 3PH, UK; 6German Federal Institute for Risk Assessment (BfR), 10589 Berlin, Germany; 7Department of Environmental Medicine, University of Southern Denmark, DK-5000 Odense, Denmark; 8Toxalim, UMR1331, INRAE, 31027 Toulouse, France; 9PrediTox, 31100 Toulouse, France; 10Altertox Academy, 1050 Brussels, Belgium; 11Biomax Informatics AG, 82152 Planegg, Germany; 12National Food Institute, Technical University of Denmark, DK-2800 Kgs. Lyngby, Denmark; 13Odense Child Cohort, Hans Christian Andersen Hospital for Children, Odense University Hospital, DK-5000 Odense, Denmark

**Keywords:** mixture risk assessment, real-life mixtures, developmental neurotoxicity, reproductive toxicity, new methodological approaches, effect-directed analysis, effect-based trigger values, PANORAMIX

## Abstract

Humans are involuntarily exposed to hundreds of chemicals that either contaminate our environment and food or are added intentionally to our daily products. These complex mixtures of chemicals may pose a risk to human health. One of the goals of the European Union’s Green Deal and zero-pollution ambition for a toxic-free environment is to tackle the existent gaps in chemical mixture risk assessment by providing scientific grounds that support the implementation of adequate regulatory measures within the EU. We suggest dealing with this challenge by: (1) characterising ‘real-life’ chemical mixtures and determining to what extent they are transferred from the environment to humans via food and water, and from the mother to the foetus; (2) establishing a high-throughput whole-mixture-based *in vitro* strategy for screening of real-life complex mixtures of organic chemicals extracted from humans using integrated chemical profiling (suspect screening) together with effect-directed analysis; (3) evaluating which human blood levels of chemical mixtures might be of concern for children’s development; and (4) developing a web-based, ready-to-use interface that integrates hazard and exposure data to enable component-based mixture risk estimation. These concepts form the basis of the Green Deal project PANORAMIX, whose ultimate goal is to progress mixture risk assessment of chemicals.

## 1. Introduction

It has been long established that chemicals from environmental and food sources, such as fluorinated chemicals, pesticides, plasticisers, flame retardants and bisphenols, can trigger adverse health effects [1]. Humans are rarely exposed to single toxicants at any given time, but rather to complex mixtures of chemicals that can combine their effects in a way that may alter their toxicity [2]. Legacy chemicals that persist in our bodies almost indefinitely, such as lipophilic persistent organic pollutants (POPs)—e.g., organochlorine pesticides, polybrominated flame retardants and per- and polyfluorinated substances (PFAS) present in everyday products, may add up with non-persistent chemicals, such as chemicals arising from pharmaceuticals, personal care products and packaging (Figure 1). Even if concentrations of some individual chemicals have decreased over the last decades as a result of management measures and the associated reductions or cessation of industrial production, the diversity of chemicals has increased, and new chemicals of concern are emerging. For example, the mixture risk quotient of certain phthalates has decreased over the last three decades, but the relative importance of the phthalate mixture effects has increased [3]. Nevertheless, the level of risk to human health arising from an everyday exposure to chemical mixtures remains largely unknown. The recent update of the Lancet commission for planetary health has stressed that “three particularly worrisome, and inadequately charted consequences of chemical pollution are developmental neurotoxicity, reproductive toxicity, and immunotoxicity” and they also referred to the relevance of mixture risk assessment (MRA) [1].

Although some advances are being made toward the incorporation of regulatory measures concerning mixture effects, current guidelines for risk assessment of chemicals are still largely grounded on the assessment of individual chemicals (Figure 1). The overall risk posed by the exposure to complex, real-life mixtures to human health is likely underestimated because (1) lack of *in vivo* and epidemiological data often prevents the derivation of a Point of Departure for toxicological endpoints and thus reduces the number of substances which can be included in an MRA, (2) MRA focuses on compounds that are relevant within the boundaries of a specific chemical legislation, but the environment and humans are co-exposed to chemicals from different regulatory silos, (3) we do not have the full picture of chemicals to which humans are exposed to, and (4) producers only need to submit individual chemical data in regulatory processes. 

To capture all chemicals that might be relevant for a MRA (industrial chemicals, biocides, pesticides, medicines, etc.), it is necessary to regulate co-exposures across regulatory silos and look at mixtures in an unbiased way. From studies on ecotoxicological effects of water quality, we know that even if we analyse hundreds of chemicals, we are still not able to explain all the toxic effects arising from these water samples by the identified chemicals [4]. This phenomenon has been observed also in *in vitro* studies on human blood and tissues, where 35 out of 100 analysed chemicals were detected, but their predicted mixture effect derived from single-compound modelling explained less than 2% of the experimental mixture effect (activation of the arylhydrocarbon receptor and cytotoxicity) [5].

Implementation of MRA across different silos is challenging from both the regulatory and scientific perspective and requires that the knowledge base on hazard and exposure covers a reasonable proportion of currently used chemicals. At present, none of our single regulatory systems alone can fully safeguard against potential risks from exposure to coincidental multi-component mixtures of substances from multiple sources via multiple exposure routes. Due to the complexity of this matter, human health assessment of aggregated exposure to multiple chemicals currently presents a great challenge to researchers, risk assessors and risk managers.

In the future, we cannot continue to rely only on data from animal studies for hazard characterisation, and the trend is moving towards using *in vitro* bioassays and so-called new approach methodologies (NAMs) as alternatives to animal testing [6,7]. Likewise, we cannot only use food intake data for exposure evaluation, as humans are often exposed to the same chemicals from many different sources, so an exposome approach that accounts for the total internal exposure might be a way forward [8].

To obtain a closer picture of the human exposome for MRA, major issues need to be addressed, such as the analytical workflow and the methodology used to monitor, identify and quantify these substances. New methodological approaches supported by advanced technologies, such as high-resolution mass spectrometry (HRMS) profiling, for suspect (SS) and non-targeted (NTS) screening, are today available to reach this goal. These innovative analytical approaches open the door to the simultaneous detection of a number of chemicals never achieved before, which is expected to become a game-changer in human chemical exposure assessment. SS aims to detect known chemicals that are expected to be present in a sample, to characterise exposure trends and to contribute to better prioritisation for further targeted developments based on generated semi-quantitative real exposure patterns. In contrast, the ambition for NTS is to identify potential unknown or new chemicals, to generate new research hypotheses and to contribute to an early warning system. Due to the diversity and complementarity of existing analytical workflows used for SS and NTS, the interpretation and comparability of the results sorted out by these approaches remains at this stage limited, and robust conclusions cannot yet be easily translated into active policies [9]. For instance, the amount of existing library databases available for data analysis and data mining in NTS is rather scarce, and only a small percentage of the exposure compounds can be framed [10]. Finally, these up-and-coming methods remain complex and require a highly technical methodological framework. We will build on methodologies developed under the EU Joint program ‘Human Biomonitoring for Europe’ to further improve the SS and NTS approaches. The mixtures considered encompass all extractable organic chemicals, no inorganics or metals unless they are organometallic compounds. Persistent organic pollutants, non-persistent organics and ionisable organics are considered.

The PANORAMIX consortium addresses this need to assess the biological impact of the combined exposure to multiple organic chemicals from different sources and, from there, perform classical MRAs as well as derive effect-based trigger (EBT) values (Figure 1). This project provides a new way of identifying “chemical mixtures of concern” and those compounds in the mixture that are mainly responsible for the mixture response (so called mixture drivers) in a diverse range of environmental, food and human samples, by creating an experimental and theoretical framework coupled with computational modelling and analysis, and a web-based chemical mixture risk calculator that can be directly used for chemical MRA and regulation.

## 2. Chemical Mixture Drivers across the Environment-Food-Human Continuum

The assessment of combined effects from realistic complex chemical mixtures based on data from *in vivo* studies alone is unquestionably limited, not only due to the huge temporal-spatial variation of co-occurring exposures and health endpoints but also due to the problematic use of test animals from an economical and ethical point of view. To circumvent this issue, a great effort has been made by regulatory agencies to incorporate non-animal based NAMs into chemical risk assessment [11,12]. NAMs comprise *in silico* and *in vitro* methods that may efficiently provide information on molecular initiating events and key events mediating adverse effects, often at a high throughput level [13,14,15,16,17].

Evidently, in a project of the scale of PANORAMIX, it is not possible to evaluate all possible mixture scenarios. Therefore, we propose a novel tiered approach where we rely on prior experience from assessing complex mixtures extracted from environmental samples and use pooled samples representative of the environment (wastewater, surface water and wild fish) as well as food (drinking and bottled water, aquaculture and wild fish and milk) in a “One Health perspective” to benchmark against existing studies [18]. We will further extend this approach to human exposure by assessing pooled blood samples from both adult volunteers and umbilical cord, as well as human breast milk. 

Regarding the exposure assessment component of PANORAMIX, samples will be collected in different European countries and pooled to approach a European ‘real-life’ mixture. For blood, pooled samples from Australia and Canada will also be included. Samples will be extracted and enriched using harmonised methods for each type of matrix, and further characterised through a combination of SS/NTS methods. Taking advantage of recent advances in chromatography coupled to HRMS, less selective strategies involving simpler sample preparation and full-scan screening have indeed emerged in the last decade to open the scope to the simultaneous analysis of suspected and unsuspected chemicals, and even the identification of unknown chemicals [19,20,21]. In addition, targeted analysis of an extended range of chemical contaminants of concern (both historical and emerging) will be performed, based on well-established sample preparation and mass spectrometric detection, identification and quantification procedures. The development and use of this integrated framework will allow the identification and/or quantification of a whole range of chemicals of the human exposome. The implementation of a set of reference standards will also generate estimated concentration data of some detected markers, leading to an integrated analysis and interpretation of the exposure effects.

Regarding the hazard characterisation component of PANORAMIX, we will apply a whole mixture-based approach, using a panel of *in vitro* bioassays with endpoints that can be mapped to classical toxicological endpoints, such as impairment of reproductive and neurodevelopmental functions. Both are known to be compromised by a great variety of chemicals, and particularly chemical mixtures [22,23,24,25]. For this purpose, each extract will be assessed in a panel of up to 20 well-established and carefully selected cell-based or cell-free *in vitro* bioassays with endpoints related to reproductive toxicity and neuro(developmental)toxicity (Figure 2). This panel includes measurement of neurotoxicity in human neuronal cells, including neurite outgrowth inhibition [26], acetylcholinesterase inhibition [27] and mitochondrial toxicity [28], neurodevelopmental assessment in zebrafish embryos [29], developmental toxicity in embryoid bodies derived from human-induced pluripotent stem cells [30], thyroid hormone disruption, such as binding to the transport protein transthyretin [31], inhibition of uptake and efflux of thyroid hormones [32], thyroid hormone receptor activity [33], activation/inactivation of thyroid hormones, iodide utilisation and recycling [32,34,35], oestrogen, androgen and aryl hydrocarbon receptor activities [36], genotoxicity, including identification of potential aneugenic and clastogenic activity [37], and finally measurement of oxidative stress [38].

Most *in vitro* bioassays used are well-established and have reached a certain maturity within their research field, including the assessment of complex environmental samples [39] as well as applications in biomonitoring [36]. Some of the assays have been standardised and validated or are expected to be validated for use in regulatory applications by Organisation for Economic Co-operation and Development (OECD) for testing single chemicals, but an extension of their regulatory application domain in mixture assessment will also be demonstrated during PANORAMIX. 

The use of *in vitro* bioassays holds great promise in providing valuable data for a MRA. However, with their increasingly regular use comes the need to better integrate all generated data and to interpret it collectively. We suggest combining our whole mixture-based approach with a component-based assessment, and the link between these two approaches will be achieved through effect-directed analysis (EDA). EDA on its own is a well-established method for the assessment of water samples [40,41,42], and its applications can be advanced and used in an innovative way through a combination with targeted screening, SS and NTS based on liquid/gas chromatography coupled to HRMS, as well as with well-informed choices of *in vitro* bioassays applied to real-life biological matrices.

In recent years, progress has been made with respect to the establishment of guidelines for SS and NTS, for instance by formulating recommendations for quality control and quality assurance of tandem mass spectral libraries [43]. These types of libraries are instrumental for obtaining high confidence annotations of HRMS data and of reliable identification of suspects. In addition, guidelines for the implementation of SS/NTS for human biomonitoring, environmental health studies and support to risk assessment were described [44]. Furthermore, to improve HRMS data annotation quality and enhance the identification success rate, different approaches for the identification of chemicals were reported, such as HaloSeeker to aid the identification of halogenated chemicals [21] and the CECscreen database, to support the annotation of chemicals of emerging concern and their (human) metabolites [45]. A workflow incorporating EDA for feature prioritisation in SS and NTS was developed and validated [46,47].

In PANORAMIX, we will apply state-of-the art EDA using a top-down approach (Figure 3), in which a relevant *in vitro* effect will guide further fractionation of mixture extracts for a non-targeted identification of chemicals (or groups of chemicals) responsible for that specific effect [36]. Importantly, EDA enables the identification of chemicals that are mixture risk drivers irrespective of their origin and use. In brief, *in vitro* bioassays will be implemented for screening of the activity in unfractionated extracts. In the case of high response, the extract will be fractionated to obtain a high-resolution bioactivity chromatogram, and HRMS data can be acquired to enable the identification of bioactive chemicals.

Our hypothesis is that the chemical drivers in a “mixture of concern” change and become more persistent and bioavailable when we move up the food chain. Using this approach, our aim is to determine how many and which chemicals drive mixture effects in humans, and show how mixtures “travel” and change, in terms of types of chemicals, concentrations and mixture ratios, from the environment to humans via the food chain.

## 3. Chemical Mixture Drivers in Human Cord Blood

Although there is concern that exposure to chemical mixtures during critical periods of development (*in utero* or during infancy) could increase the risk of neurodevelopmental and reproductive disorders, researchers have primarily studied this by component-based mixture approaches to determine if and how the chemicals interact. However, this approach is usually limited to a small number of compounds, and therefore can provide only limited insight into the association between human health effects and combined exposure to chemicals. 

As one possible strategy, we suggest characterising the link between mixture exposure of the foetus and the development of reproductive and/or neurobehavioural dysfunction at early age. With this purpose in mind, we will analyse cord blood samples collected at birth in the Odense Child Cohort (OCC) (Figure 4). OCC is an on-going prospective birth cohort wherein newly pregnant women residing in the Odense Municipality were offered participation from 2010–2012 [48]. Following enrolment, the women’s general health and lifestyle was assessed. Gestational age of the mothers, as well as birth weight, length and head and abdominal circumference of the newborns were obtained from birth records and a cord blood sample was stored. At child age of 3 months, 1½, 3, 5 and 7 years, parents answered questionnaires on child and maternal health, and the child underwent a clinical examination. Currently, over 2500 children are enrolled in this cohort and data on important health outcomes for the children is well-documented, in particular reproductive toxicity-related outcomes, including anogenital distance as a non-invasive hallmark of incomplete masculinisation and several sex hormones, as well as neuropsychological development function, such as the assessment of language development (vocabulary and complexity) using MacArthur-Bates Communicative Development Inventories report, the Child Behaviour Checklist for ages 1½–5 years that comprises 100 questions regarding behavioural, emotional, and social problems, six of which represent an attention deficit hyperactivity disorder problem scale, and the intelligence quotient score at the age of 7, based on subtests of the original Wechsler’s Intelligence Scale for Children. Thus, this cohort can provide unique cord blood samples to study chemical mixture drivers to which the foetus is exposed.

PANORAMIX will apply developed methodology and scientific evidence provided by the first phase of the project, from which 3 to 6 bioassays, performing best in terms of specificity, selectivity and sensitivity, will be selected to assess mixture effects of extracts from 500 cord blood samples from the OCC (Figure 2; Phase 2). Resorting to chemical profiling coupled to EDA, we will identify and quantify chemical mixture drivers in children and associate this data with existing health information and measures of adverse effects in the children collected from the OCC (Figure 4).

We hypothesise that with this approach, we will identify several chemicals of emerging concern that elicit specific *in vitro* effects and can be associated with specific disorders which so far have flown under the radar.

Furthermore, by involving samples from this well-characterised child cohort, PANORAMIX will take the first steps toward associating adverse health outcomes to chemical mixture exposures in early life.

## 4. State of the Art of Mixture Risk Assessment

Mixture toxicity as such, i.e., the ability of chemical mixtures to cause more harm to human health than their individual constituents under certain circumstances, is broadly accepted given the numerous reports of mixture effects under laboratory *in vitro* and *in vivo* conditions. However, there is still a long way to go regarding the implementation of regulatory actions on MRA [49,50,51].

Guidelines on health risk assessment of chemical mixtures have been establish by the United States Environmental Protection Agency (USEPA) for over 30 years [52]. In Europe, options for estimating maximum residue levels were primarily discussed by the European Food Safety Authority (EFSA) with regard to pesticides [53]. More recently, considerations on different approaches for MRA were shared by both EFSA [54] and the OECD [55]. These documents provide guidance on both component-based and whole mixture-based approaches. In the latter approach, a MRA is performed from a more holistic point of view, where each mixture as a whole is evaluated in the same way as an individual chemical is.This whole-mixture MRA can be used for the assessment of poorly defined mixtures, as is the case of the so-called UVCB substances, i.e., ‘Substances of Unknown or Variable Composition, Complex Reaction Products and Biological Materials’ [56]. Its main advantage is that this MRA approach reflects the integrated contribution of all chemicals within a given mixture, even when it comprises unknown or unidentified substances, as well as potential interactions among them. However, it does not provide information on the toxicity of individual mixture components and does not allow the identification of the chemicals that drive mixture effects. Moreover, it is grounded on the assumption that the mixture composition (quantitatively and qualitatively) does not vary significantly over time or across individuals [57]. 

On the other hand, the component-based approach accounts for the compositional variability of the mixtures, predicting risk based on individual chemicals present in a mixture, and it is currently applied by many risk assessment frameworks [58,59,60]. For this approach, closely related chemicals are considered as a group, rather than as individual substances, being assigned to cumulative assessment groups (CAGs) based on individual toxicological properties. In Europe, this classification was first recommended by EFSA for the risk assessment of pesticides, with thyroid and nervous system as common target organs [61], and recently this recommendation was further expanded to other chemical classes and other endpoints [62]. Several CAGs have been proposed so far, including common mechanisms or modes of action, or common adverse outcome pathways [63,64,65,66]. This categorisation has allowed a component-based MRA to be established on the default assumption of dose addition, meaning that the individual hazard quotients of known chemicals within the same CAG are simply added up to a hazard index (HI) [67,68]. For example, using this component-based MRA approach, it was reported that the combined exposure to four polybrominated flame retardants, with neurodevelopmental adverse effects in common, had an estimated hazard index that exceeded acceptable levels in breastfeeding infants and small children, even at moderate exposure levels [69]. 

Similarly, in a study by Kortenkamp et al. [70], the cumulative risk from 29 antiandrogenic substances from different chemical groups, including plasticisers, bisphenols, pesticides and preservatives on semen quality was estimated to surpass tolerable levels for people on the upper end of exposure levels by 100-fold. Bisphenols, polychlorinated dioxins and paracetamol were found to drive risks to semen quality. Such studies highlight the need to limit human exposure to chemicals that might be hazardous when combined in real-life mixtures, even when they are individually present below their regulatory thresholds. 

Mixture responses for compounds from the same CAG have been shown in many experimental studies to be accurately predicted by dose addition. For instance, the foetal testosterone production affected by a chemical mixture was accurately predicted with the dose addition model, based on available data of individual chemicals [71]. In this study, mixture effects of five antiandrogenic phthalates were predicted from individual data for 11 out of 14 endpoints of abnormal postnatal male reproductive tract development in rats. Orton et al. [72] showed that the combined antiandrogenic effects of 30 substances, from a wide range of chemical groups, occurred even at very low concentrations of individual components, with concentration-response curves closely following dose addition estimations.

Nevertheless, and although rarely observed in experimental studies, the dose addition model can underestimate combination effects. For instance, Christiansen et al. [73] showed that a mixture of four antiandrogenic chemicals (one phthalate, two fungicides and one pharmaceutical) with different modes of action, composed and tested at their no observed adverse effect levels (NOAELs), agreed well with the dose-addition prediction for most hallmarks of disrupted male sexual development (anogenital distance (AGD), retained nipples and sex organ weights), but underestimated the malformations of external sex organs. Experimental synergistic effects have been reported mainly at high effect levels in combination with binary or ternary mixtures [74]. The competing case of overestimating mixture responses (antagonism) and for which the dose addition assumption may be too conservative, appear to be even less frequent [75], and are often reported for specific groups of chemicals like metals [76,77,78]. Here, the pharmacological prerequisite of the dose addition model, i.e., all compounds in the mixture share a “similar” pharmacological site or mode of action, is often not fulfilled. In the most extreme case, all compounds might act on distinct molecular target sites, which would suggest the independent action mixture model as the better reference. At environmentally realistic effect levels below 10% effect, the difference between mixture predictions of dose addition and independent action becomes negligible as the number of chemicals in a mixture increases [79].

It is commonly accepted that synergisms are more likely to occur with an increasing complexity of the endpoint, i.e., simple receptor activation endpoint offer *a priori* less possibilities to deviate from the dose addition assumption than a functional *in vitro* endpoint or apical *in vivo* endpoint. 

Importantly, a comprehensive component-based MRA frequently requires the assignment of chemicals from different classes into the same CAG, and this categorisation still remains a challenge. We suggest overcoming this issue by resorting to specific *in vitro* data of chemicals, where active chemicals by default can be considered to belong to the same CAG. This approach assumes that chemicals are acting additively within a mixture, that the *in vitro* effect can be extrapolated to an *in vivo* effect, and that the molecular initiating event or key event translate to an adverse outcome *in vivo*, meaning that the chemicals are part of the same CAG. We will challenge this hypothesis by comparing the overlap between compounds of a CAG derived by the proposed *in vitro* classification with the target organ-based approach suggested by EFSA [62].

## 5. Pragmatic Solutions for Mixture Risk Assessment

### 5.1. Designed Mixture Experiments for Assessing Alignment with or Deviation from Additivity

Generally, it is accepted that chemicals can be grouped for MRA according to modes of action, not according to chemical structural similarity or use/function-based criteria. Chemicals with a similar mode of action are assumed to act dose-additively at low effect levels in, while synergism and antagonism may play a minor role in environmentally realistic mixtures and mixtures that we find in the human body [74,75]. However, it remains to be evaluated whether deviations from additivity can occur in mixtures of a large number of compounds at realistic exposure concentrations in blood for various bioassay endpoints. Thus, it remains to be determined whether dose addition affords sufficient protection. 

For this purpose, we will design complex mixtures containing up to 50 chemicals, based in the concentration ratios from human blood samples, and model and predict the mixture effects on reproductive function and developmental neurotoxicity with the aim of determining if there are any major deviations from additivity at realistic exposure levels or if dose addition is a robust prediction model for mixtures in human blood. 

### 5.2. Case Studies for Evaluating Mixture Assessment Factors and Safety Margins

Instead of assessing risk from mixtures, a pragmatic, intermediate measure might be to lower safety limits for single chemicals by (a) certain factor(s), so-called mixture assessment factors (MAF). To provide input on the general need for and magnitude of evidence-based MAFs for implementation by regulators, PANORAMIX will conduct knowledge-building theoretical case studies on mixture effects of potential reproductive and developmental neurotoxicants. These case studies will allow the identification of mixture drivers, the evaluation of current safety margins (as provided by the substance-by-substance approach), and the recommendation of potential case-specific MAFs. 

### 5.3. Derivation of Effect-Based Trigger Values

Over time, *in vitro* bioassays, in particular reporter gene assays, have been tuned to be sufficiently sensitive to detect effects in enriched samples of clean waters, such as drinking water and recycled water [39]. This high sensitivity means that a positive result in a bioassay does not necessarily mean that there is concern for related real-life exposure. To help distinguish between an acceptable and an unacceptable bioassay response, the establishment of EBT values has been proposed [79]. EBT values have been shown to adequately set a quality threshold for different types of water, including drinking water [80,81]. However, they have never been established for assessing mixture effects in food or human samples. 

In an analogy to the successful application of the EBT concept in water quality assessment, PANORAMIX aims to develop mechanism-based EBT values derived from endocrine disruption and neurotoxicity bioassay results performed on food and human blood samples.

Through this innovative experimental approach, we will derive EBT values for mixtures in the environment, food, and humans that can be used for determining if a mixture in a sample is hazardous or not and for setting acceptable limits for mixture exposures.

### 5.4. Integrative Web-Based Chemical Mixture Calculator

To fulfil the need for a tool for MRA, we will develop a flexible web-based interface comprising hazard and exposure data for a large number of chemicals which—based on the assumption of dose additivity—can be applied for an initial assessment of risk from mixtures of chemicals defined by the user.

A basic prototype of the Chemical Mixture Calculator (CMC) has been built [68] and presently contains 200–300 data-rich compounds. The CMC allows the estimation of hazard indices based on the assumption of dose addition for seven types of adverse effects. The current platform will be further populated by *in vitro* and exposure data from the chemical mixture identification and bioassay testing and will incorporate existing datasets from the ICE (https://ntp.niehs.nih.gov/whatwestudy/niceatm/comptox/ct-ice/ice.html, accessed on 1 June 2022), CompTox (https://comptox.epa.gov/dashboard/, accessed on 1 June 2022) and IPCHEM (https://ipchem.jrc.ec.europa.eu/, accessed on 1 June 2022), to ensure that the 2.0 version of the CMC integrates newly available hazard and exposure data. The underlying hypothesis is that risk characterisation ratios or hazard quotients can be calculated based on toxicokinetic-corrected *in vitro* data for key events and human biomonitoring data.

In the end, the intention is to deliver a user-friendly, integrative solution, ready for use by the widespread scientific community and by risk assessors and regulators, which will allow instantaneous MRA for both data-rich and data-poor substances. 

## 6. Conclusions

With our outlined workplan, we will contribute to four major goals: (1) provide scientific evidence to enable prevention and/or mitigation of co-exposure to environmental chemicals in the environment and in humans; (2) support the implementation of existing risk assessment and risk management approaches to reduce the most critical exposures, including the setting of EBT values for mixture effects from bioassays; (3) evaluate new regulatory approaches, such as the MAFs; (4) support activities on combined exposures as relevant for the ‘Strategic Approach to Pharmaceuticals in the Environment’ and as to be defined in the forthcoming implementation of the ‘Chemical Strategy for Sustainability’.

## Figures and Tables

**Figure 1 ijerph-19-12990-f001:**
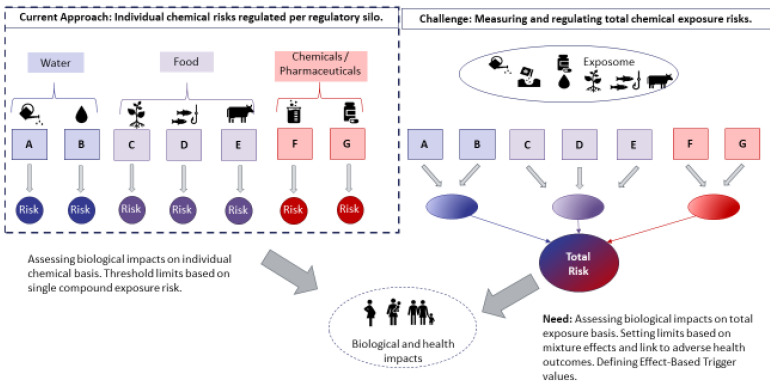
Current approach on chemical risk assessment and the challenge of mixture risk assessment addressed by PANORAMIX. The letters A–G refer to different compounds (modified from [2]).

**Figure 2 ijerph-19-12990-f002:**
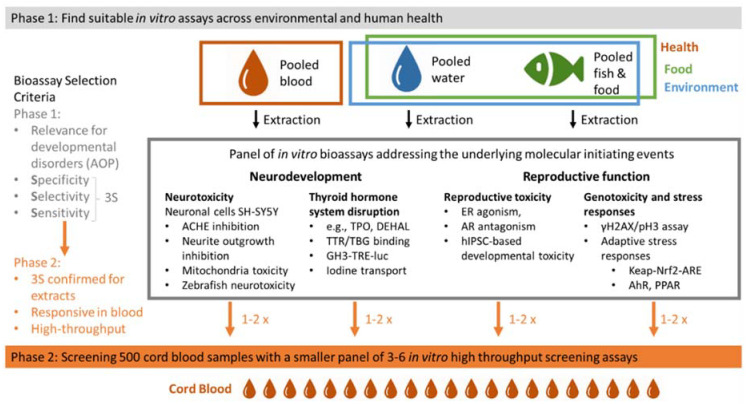
Overview of Phases 1 and 2 of the workplan for whole mixture screening based on high-throughput in vitro new methodological approaches. Pooled blood/water/fish/food: samples from a large number of European countries or European individuals are pooled, and thereafter the chemicals are extracted from the pools. ACHE: acetylcholinesterase; TPO: thyroid peroxidase; TTR: transthyretin; TBG: thyroid hormone binding globulin; GH3-TRE-luc: reporter gene bioassay that can measure thyroid hormone receptor mediated activity; ER: estrogen receptor; AR: androgen receptor; hIPSC: human-induced pluripotent stem cells; Nrf2: nuclear factor erythroid 2–related factor 2; AhR: arylhydrocarbon receptor; PPAR: peroxisome proliferator-activated receptor.

**Figure 3 ijerph-19-12990-f003:**
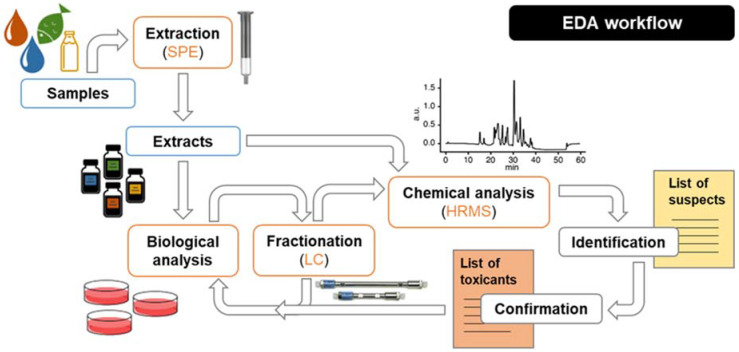
Schematic representation of the workflow entailed by the effect-directed analysis approach. EDA: effect-directed analysis; HRMS: high resolution mass spectrometry; LC—liquid chromatography; SPE: solid-phase extraction.

**Figure 4 ijerph-19-12990-f004:**
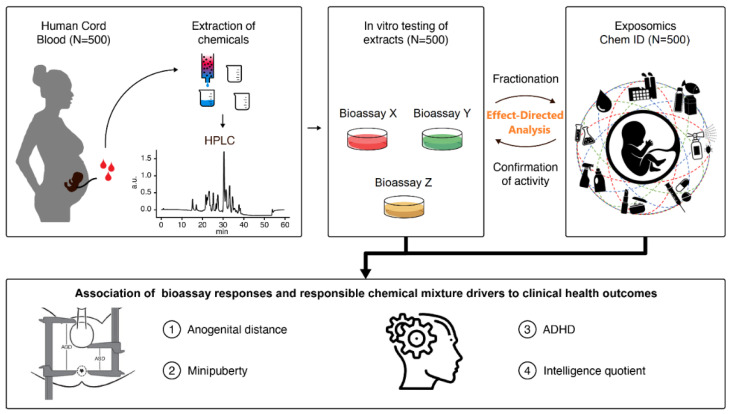
Illustration of the approach for the identification of a potential correlation between whole mixture-based in vitro outcomes and identified chemical mixture drivers in human cord blood samples and reproductive and neuropsychological health outcomes in children. ADHD: attention deficit hyperactivity disorder.

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
