# Peer review of "Mixture Risk Assessment of Complex Real-Life Mixtures—The PANORAMIX Project"

_ijerph, 2022, doi:10.3390/ijerph192012990_

Round 1

Reviewer 1 Report

The paper addresses the emerging problem of exposure of the general population to mixtures of many chemicals. The paper is well written, but I have a few comments and suggestions.

1. The Authors presented in coclusion goals of the project. According to that, results will only provide essential data for risk assesment, not risk assesment itself. Therefore, the Authors should consider changing the title.

2.The authors could briefly state what substances will be included in the study. Only organic, such as POPs? Will the Authors include metal or non-metal compounds (mercury or lead)?

3. Would another extraction method be used, or only SPE? Would pre-isolation of fat from the matrix be included for POPs analysis?

Author Response

We thank you for the feedback from the reviewer. Our responses are included below.

Rewiever 1:

  1. The Authors presented in conclusion goals of the project. According to that, results will only provide essential data for risk assessment, not risk assessment itself. Therefore, the Authors should consider changing the title.

We agree and will suggest the title: ‘Mixture Risk Assessment of complex real-life mixtures – the PANORAMIX project’, which is now included in the manuscript.

  1. The authors could briefly state what substances will be included in the study. Only organic, such as POPs? Will the Authors include metal or non-metal compounds (mercury or lead)?

The project will focus on the development, consolidation and application of complementary strategies for large-scale screening of markers of exposure from various substance classes including pharmaceuticals, plasticizers (phthalates, bisphenols), UV-filters, pesticides, and perfluorinated alkylated substances (PFAS). The applied strategies will also cover a range of persistent organic pollutants (POPs) of emerging concern that are also an issue with regard to some health outcomes covered by the PANORAMIX project (reproductive and neurobehavioral functions).

The mixtures considered encompass all extractable organic chemicals, no inorganics or metals unless they are organometallic compounds. Persistent organic pollutants as well as non-persistent organics as well as ionisable organics are considered. Generally, metal and inorganics analysis is 99% of the time performed using targeted analysis, and these are not real candidates in an effect-directed analysis.

The underlined text is now mentioned in the paper L. 128.

  1. Would another extraction method be used, or only SPE? Would pre-isolation of fat from the matrix be included for POPs analysis?

The developed sample preparation will be basically non-selective, in line with the broad range of exposure markers expected to be simultaneously captured. Water and blood will be extracted with established SPE methods using sorptive materials that enrich polar and non-polar as well as neutral and ionised organic chemicals.

Although a de-lipidation step is envisaged to discard most part of the interfering endogenous lipids that may disturb the further analysis, a liquid-liquid partitioning using both polar and non-polar solvent will be applied for milk and fish samples, permitting to preserve the possibility of monitoring both polar/non-persistent and more lipophilic/persistent organic chemicals, these two resulting fractions being analyzed respectively through LC-HRMS and GC-HRMS.

Reviewer 2 Report

Manuscript Number: ijerph-1897373 

Title: Providing Risk Assessments of Complex Real-Life Mixtures for the Protection of Europe´s Citizens and the Environment - the PANORAMIX Project

The paper provides suggestions for risk assessments of complex mixtures in the real life, which helps to progress mixture risk assessment of chemical. The paper is good, I just have some minor comments to improve the paper, see below:

1.     Line 44:Where is (3)?

2.     Figure 2: What is pooled blood?

3.     The paper wants to assess the biological impact of the combined exposure to multiple organic chemicals from different sources, what about inorganics? Both organics and inorganics exist in the environment.

4.     Is blood a good biomarker for all inorganics?

Author Response

Rewiever 2:

 Line 44:Where is (3)?

Unfortunately, we do not understand this question.

2. Figure 2: What is pooled blood?

We mean that we pool blood from a larger number of individuals and thereafter extract the chemicals from the pool. This is now mentioned in the figure legend.

3. The paper wants to assess the biological impact of the combined exposure to multiple organic chemicals from different sources, what about inorganics? Both organics and inorganics exist in the environment.

The number of inorganic and metal compounds is very limited and sum parameters such as total element concentrations are readily available, so metals and organics can be easily evaluated with the traditional chemical analytical methods such as ICP-MS. In contrast the abundance and diversity of organic chemicals is overwhelming and chemical analysis alone can never capture all chemicals on the market, let alone their human and environmental transformation products. Therefore, the focus is on the mixture effects of those known and unknown chemicals.

4. Is blood a good biomarker for all inorganics?

Not relevant, as we do not measure inorganics.